# Social and nonsocial synchrony are interrelated and romantically attractive
M. Cohen ⓘ , M. Abargil ⓘ , M. Ahissar ⓘ & S. Atzil ⓘ ✉

The mechanisms of romantic bonding in humans are largely unknown. Recent research suggests that physiological synchrony between partners is associated with bonding. This study combines an experimental approach with a naturalistic dating setup to test whether the individual differences in social and nonsocial synchrony are interdependent, and linked to romantic attractiveness. In a preregistered online experiment with 144 participants, we discover that inducing physiological synchrony between an actor and an actress determines their attractiveness ratings by participants, indicating that synchrony can increase perceived attraction. In a lab-based naturalistic speed-dating experiment, we quantify in 48 participants the individual tendency for social physiological synchrony, nonsocial sensorimotor synchrony, and romantic attractiveness. We discover that the individual propensity to synchronize in social and nonsocial tasks is correlated. Some individuals synchronize better regardless of partners or tasks, and such *Super Synchronizers* are rated as more attractive. Altogether, this demonstrates that humans prefer romantic partners who can synchronize.

Romantic bonding is a central feature of human life. It is associated with well-being[1–3], physical health[4–10], mental health[11–13], happiness[14], life satisfaction[15], development[16], and sexual desire[17]. Yet, the neural and behavioral mechanisms that determine why we are attracted to selected individuals, and not others, are still unknown. From an evolutionary perspective, attractiveness reflects a preference for adaptations that increase survival and reproduction[18,19]. Accordingly, attractiveness is affected by physical features[20–22], resources, social position, and strength[23–26].

In addition to such static features of fitness, recent research points out that dynamic interactive features, such as synchrony, may also determine attraction in humans. We and others recently demonstrated that physiological synchrony predicts romantic and sexual attraction between partners during a speed-date[27,28]. In psychobiology, dyadic synchrony refers to the temporal matching of rhythms in physiology, behavior, or affective states between two partners[29]. Synchrony between humans has been widely documented across different measures and time scales, including neural function[30–35], arousal[36], respiration[37–41], heart rate[40,42–46], hormones[47–50], motion[34,51–53], and behavior[46,54–57]. Across domains, physiological synchrony between adults is consistently associated with social cooperation[58], romantic satisfaction[39], and sexual satisfaction[59,60]. In addition to adult-adult bonds, synchrony was studied in parents and infants as a mechanism for bonding, attachment, and co-regulation[54,61–67], predicting developmental outcomes and social skills[68]. Altogether, the ability to synchronize with a partner is a central feature of social behavior.

Despite its key role in social behavior, synchrony is not exclusively a social phenomenon, as synchronization is evident not only between individuals but also within individuals, such as in sensorimotor control[69–72]. While social and nonsocial synchrony have been widely studied, it is still unclear whether they share common mechanisms[73–76]. Interestingly, recent research discovered that individuals with autism show impaired performance in a sensorimotor synchronization task, where they are requested to adjust a finger tap to synchronize with an external metronome beat[71,72], compared to neurotypicals[77]. Moreover, sensorimotor synchronization was associated with the self-reported score of social function[77]. Based on this, we hypothesize that the ability for social interactions is rooted in domain-general features of sensorimotor integration, which is key for both social and nonsocial synchrony.

This research aims to answer two key open questions about the role of synchrony in romantic attraction. First, does synchrony have a role in eliciting attraction? Alternatively, synchrony can result from an increased attraction that raises the motivation to synchronize. Such interactions can also be bi-directional. Second, are there individual differences in the ability to synchronize that determine attractiveness? To address these questions, we ran two experiments. The first is an online experiment that tested whether experimentally manipulating the level of synchrony between a man and a woman determines their attractiveness ratings by 144 participants. The second is a speed-dating experiment (Fig. 1), in which participants meet for four speed-dates with different partners in a round-robin setup (48

The Hebrew University of Jerusalem, Jerusalem, Israel.
✉e-mail: shir.atzil@mail.huji.ac.il

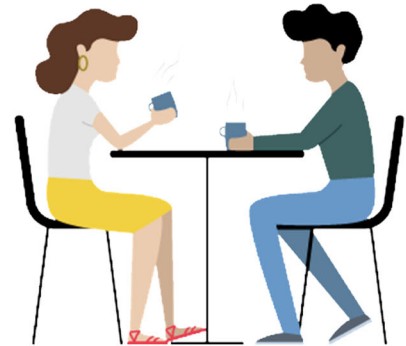 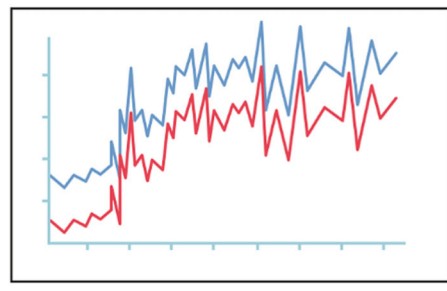

**Fig. 1 | The experimental setting of the speed dating lab experiment.** Forty-eight men and women meet for five-minute speed dates in a dedicated room with a homey arrangement while their physiology is sampled at 4 Hz using Empatica E4 wristbands[28]. Physiological synchrony between partners is calculated as the Pearson Correlation between the levels of electrodermal activity (EDA) of the man and the woman during the date. Participants rate their interest in the partner twice, once before they start to interact, and again after the date is complete. To assess individual rates of synchrony and attraction, participants partake in four dates, and we calculate their *Individual Electrodermal Synchrony Scores* and their *Individual Romantic Attractiveness Scores* by averaging the scores across their dates. We investigate whether initial interest affects physiological synchrony during the date and whether individual synchrony scores predict individual attractiveness scores.

participants, 85 dates). We quantified in each participant the individual levels of *social synchrony*, or tendency to synchronize regardless of the partner, by calculating their average electrodermal synchrony with different partners during speed-dates. In addition to social synchrony, we quantified the individual performance of *nonsocial synchrony* in a finger tapping sensorimotor task. Last, we calculated the individual scores of *romantic attractiveness* by averaging the attractiveness scores each participant received from their different partners. We tested whether there are *Super Synchronizers* who can better synchronize regardless of the task or the partner and whether Super Synchronizers are rated as more attractive.

## Methods

### Online experiment

**Sample size and power analysis.** Before the experiment, we estimated the target sample size using power analysis (G*Power 3.1.9.4) for repeated measures MANOVA between two different groups, with one independent variable (synchrony) and two dependent variables (attractiveness of the actors, attraction between the actors). The parameters of the power analysis are a power of 0.95 to detect an effect size of Cohen's f = 0.3 at a standard α = 0.05 error probability rate, with an estimated correlation between repeated measures of r = 0.8. The target sample size was 132 participants (66 participants per condition), and we recruited 160 participants. This experiment was preregistered on August 28, 2022 – https://doi.org/10.17605/OSF.IO/H9WYR.

**Participants.** One hundred sixty participants were recruited via iPanel to participate in the online experiment. Out of these, 16 participants were excluded for not meeting the inclusion criteria (15 reported a diagnosis of a psychiatric disorder, and one exceeded the predefined age range). The final dataset comprises 144 participants (76 women; gender was determined based on participant identification), aged 18–30 (*Mean* = 24.89, *SD* = 3.61 years). Participants were randomly assigned to one of the two conditions: watch a high synchrony interaction (73 participants); or watch a low synchrony interaction (71 participants). All participants were Hebrew speakers. Participants were remunerated for their participation. The ethical committee of the Faculty of Social Sciences of the Hebrew University of Jerusalem approved the experiment per relevant guidelines and regulations. Each participant signed an informed consent form before participation.

**Procedure.** In the first stage of the experiment, participants completed a socio-demographic questionnaire. Then, participants watched a 92-second video of a low or high synchrony interaction between an actor and an actress. After the video, participants provided four attractiveness ratings[78–81]: (1) how romantically attractive is the man in the video, (2)

how romantically attractive is the woman in the video, (3) the attraction of the man in the video to the woman in the video, (4) the attraction of the woman in the video to the man in the video. At the end of the experiment, participants were asked to rate the perceived behavioral synchrony on the date as a manipulation validation. This question was introduced at the end of the experiment to avoid priming the participants towards the phenomenon of synchrony while rating the attractiveness of the actors. The t-test between the perceived synchrony ratings for the synchronous condition (*Mean* = 5.863, *SD* = 2.411) and the non-synchronous condition (*Mean* = 4.338, *SD* = 2.443) shows a significant difference between the two conditions (t = −3.769, df = 141.76, p-value < 0.001, *Cohen's d* = 0.628, 95% *Confidence Interval* = [0.966, 0.291]). Attractiveness and synchrony ratings were rated on a Likert scale (0 is the lowest score, 10 is the highest score) using Qualtrics.

**Stimuli.** The stimuli included videos of a man and a woman on a date performed by actors. The same actor and actress participated in both conditions (high synchrony interaction; low synchrony interaction), and the manipulated variable in the videos was the level of social synchrony between the actors. This was achieved by directing instructions given to the actors during filming. In the synchronous condition, the actors were directed to increase their physiological synchrony by being more attuned and sensitive to the partner and trying to adapt to them. In the non-synchronous condition, the actors were directed to lower their physiological synchrony by acting independently of their partner, being less affected by them. The videos were controlled for the content of the actors' conversation, the physical appearance of the actors, the settings (a dedicated room at our lab with a homey arrangement), the actors' positions, and the length of the videos (92 seconds). The level of synchronization in the videos was confirmed with physiological assessments of electrodermal activity during the interactions. In the high synchrony interaction, the electrodermal synchrony was r = 0.614, compared to r = −0.097 in the low synchrony interaction. To further address potential differences between the two videos, we conducted a post hoc behavioral analysis to characterize the behavioral display of the actors in the two videos (Supplementary Note S1, Supplementary Table S1, and Supplementary Fig. S1).

**Physiology.** During each interaction, we quantified the electrodermal synchrony between the partners by measuring the electrodermal activity of each person using Empatica E4 wristbands placed on the wrist of the left hand. Electrodermal activity refers to the continuous variation in the electrical characteristics of the skin. Varying numbers of eccrine sweat glands secrete varying amounts of sweat, depending on the degree of sympathetic activation, and as more sweat is being secreted,

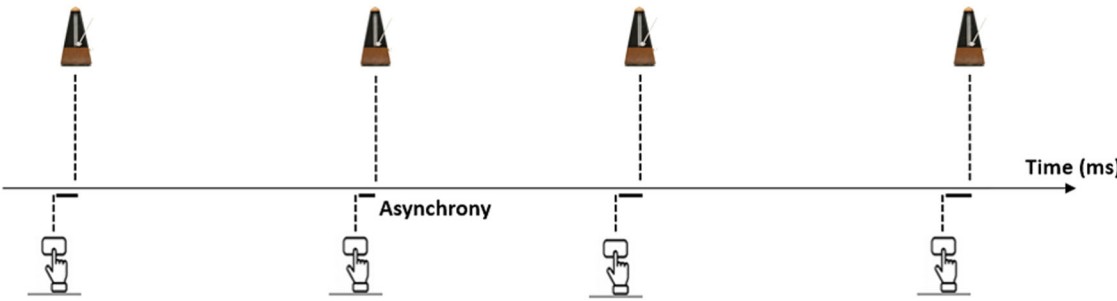

**Fig. 2 | The temporal structure of the finger tapping task.** Participants completed a tapping task, where they were asked to synchronize their finger tap on a wooden box to the beat of an external metronome[71,72,77]. The illustrations of the metronome represent the timing of the auditory beats, and those of the hand represent the participant's responses. Perfect sensorimotor synchrony reflects perfect alignment between the auditory beats and the participant's taps, whereas more significant gaps represent lower synchrony. Off-beat errors are typically negative because participants' tapings usually precede the beat, demonstrating that they learn to predict it. *Individual Sensorimotor Synchrony Scores* were calculated as the average size of the off-beat temporal gap (i.e., asynchrony) between the auditory beats and participant's taps across the entire task.

electrodermal activity increases[82–84]. Typically, electrodermal activity is measured as skin conductance by applying a small, constant voltage to the skin. Skin conductance can be calculated by measuring the current flow through the electrodes, as the voltage is kept constant[83,85,86]. Electrodermal activity is controlled by the sympathetic nervous system and reflects the level of physiological arousal[82,87]. Electrodermal activity is a validated and standard measure in psychology[82,88]. Furthermore, electrodermal activity has been reported to be sensitive to social stimuli and reactive during social interactions[89,90], specifically between romantic partners[27,28,36], making it a valid autonomic measure for social synchrony in a romantic context.

The wristbands contain an electrodermal activity sensor with a sampling frequency of 4 Hz, resolution of one digit –900 pSiemens, range of 0.01–100 μSiemens, and alternating current (8 Hz frequency) with a maximum peak to peak value of 100 μAmps (at 100 μSiemens) (https://empatica.app.box.com/v/E4-User-Manual). The obtained electrodermal signal was uploaded to the E4 application for Windows after each run, then downloaded from the Empatica website for processing and analysis. Previous studies reported using Empatica E4 wristbands in behavioral experiments that measured electrodermal activity[91–93]. Notably, the E4 wristband is quick to connect, wearable, and wireless, hence it does not interfere with natural behavior in experimental settings, enhancing the ecological validity of the obtained results (https://empatica.app.box.com/v/E4-User-Manual, https://empatica. app.box.com/v/E4-getting-started). The E4 device was validated in a previous dating experiment conducted in our lab[28], which was consistent with additional validating studies[94–96]. The preprocessing of the raw data included the temporal alignment of the data from both partners based on a global timestamp (https://empatica.app.box.com/v/E4-User-Manual, https://empatica.app.box.com/v/E4-getting-started). Then, to calculate the electrodermal synchrony between the partners, we applied Pearson Correlation using R (2022.12.0 + 353)[28].

### Speed-dating experiment

**Sample size and power analysis.** Power was calculated according to the association between synchrony and romantic attractiveness. Given the multiple sources of random variability in a design, simulations that capitalize on random effects revealed in actual data are the most accurate method of determining the power[97]. Hence, to estimate our target sample size, we used bootstrapped power calculation on a separate dataset of a previous speed-dating experiment from our lab with 30 participants[28]. Using bootstrap sampling ($n = 10,000$) to assess the power of different sample sizes, we found that to reach a statistical power of 95%, a sample size of 48 participants is needed. Forty-eight participants were recruited, out of which physiological data were available for 32 participants (64

dates), with a statistical power of 81% in the analyses that include electrodermal synchrony. This experiment was not preregistered.

**Participants.** Forty-eight students (24 men and 24 women; gender was determined based on participant identification), aged 19–28 (*Mean =* 24.72, *SD =* 1.93 years) participated in the speed-dating experiment. The experiment was conducted over seven runs, where in each run, four men and four women were invited to the lab to meet in a speed-dating round-robin rotation. Of the forty-eight participants, thirty-two completed dates with all four partners. Due to last-minute participant cancellations, ten participants completed dates with three partners and six participants with two partners. Due to equipment malfunction, the physiological data was not successfully collected for runs 5–7 (16 participants; 8 women), and thus, these runs are not included in the physiological analyses. Participants were recruited via social media and the university's online experiment system. All participants are native Hebrew speakers, not diagnosed with any psychiatric disorder, heterosexual, cis-gender, single, and interested in a romantic relationship. The ethical committee of the Faculty of Social Sciences of the Hebrew University of Jerusalem approved the experiment per relevant guidelines and regulations. Each participant signed an informed consent before participating and was remunerated for their participation.

**Procedure.** Men and women met in a speed-dating round-robin rotation. Each date lasted five minutes and took place in a dedicated room with a homey atmosphere. At the beginning of each date, participants reported their initial interest by rating their interest in succeeding on this specific date on a scale of 1–5, where one is not interested and five is highly interested. Immediately after each date, participants rated their post-date attraction to the partner. We recorded the participants' electrodermal activity during the dates.

After the dates, participants completed a paced finger tapping task. The participants were asked to synchronize to the beat of an external metronome by tapping with their finger on a wooden box[71,72,77] (Fig. 2). The task was composed of six blocks, using three protocols (two blocks of each): fixed metronome tempo (2 Hz, beat every 500 ms); tempo alternated (every random 7–13 intervals) between two beats: in one, they differed by 120 ms; in the other, by 180 ms. Each block lasted sixty seconds, with a five-second break before each block (six minutes and thirty seconds total).

**Electrodermal synchrony Scores.** Synchrony reflects partners' simultaneous alignment of electrodermal activity, and was calculated for each date as the Pearson Correlation between the two partners' electrodermal activity during the date using R (2022.12.0 + 353). Given that every participant partook in multiple dates, the electrodermal synchrony

scores were averaged per participant across all of their dates to compute an *Individual Electrodermal Synchrony Score*. Our previous research demonstrated that synchrony during the first two minutes of the date is primarily predictive of the romantic attraction between the participants[28]. Thus, synchrony was calculated during the first two minutes and across the entire date; see Supplementary Notes S2.1, S2.2, and S2.3 for all electrodermal synchrony measures. Bonferroni correction is applied to control for multiple hypothesis testing.

**Romantic attractiveness scores.** Attractiveness reflects the attraction ratings that each participant received from their partners. After each date, participants rated their level of attraction to the partner on a scale of 1–5. The romantic attraction ratings of each participant were averaged across all of their dates, computing an individual measure of *Individual Romantic Attractiveness Scores*. Out of 85 dates, romantic attraction scores were successfully collected on 83 dates. On one date, neither partner completed the attraction questionnaire, and on another date, the man did not complete it.

**Sensorimotor synchrony scores.** The finger tapping task quantifies how accurately and reliably participants synchronize their finger tap to an auditory metronome beat. Perfectly synchronous behavior means perfect alignment between the auditory beats and the participant's taps, whereas more significant gaps represent lower synchrony. Each participant is given an *Individual Sensorimotor Synchrony Score* based on their mean performance score on the task. The score is calculated by averaging the size of the off-beat temporal errors in milliseconds between the auditory beats and participant's taps (referred to *asynchrony*) across the 6-minute duration of the task. A perfect performance is scored 0, representing complete sensorimotor synchrony with no off-beat temporal error. In contrast, increased negative values (where the finger taps typically precede the metronome beats) represent poorer sensorimotor synchrony between the auditory stimuli and motor finger taps. Sensorimotor data collection was completed for 44 participants.

**Testing the association between the initial interest and electrodermal synchrony during the date.** We tested whether initial interest in the partner predicts the rate synchrony during the date. Since this analysis is performed on separate dates, we applied a multilevel model to account for the statistical non-independence of the data points as similar individuals participated on four dates, nested in four different runs[28,98,99]. The initial interests of both the man and the woman were applied as fixed effects, while accounting for the random effects for the intercept of recurring data from individuals that repeated on different dates and for their nesting in specific runs. To augment the classical statistical inference, we also included Bayesian analyses and computed Jeffreys-Zellener-Siow (JZS) Bayes Factors (BFs). The default prior settings (used by R) were left unchanged. $BF_{01}$ values around 3 provide weak to moderate support in favor of the null hypothesis[100]. The analyses include 64 dates between thirty-two participants (16 men, 16 women), aged 19–28 (*Mean* = 24.8, *SD* = 2.02 years).

**Testing the association between electrodermal synchrony scores and sensorimotor synchrony scores.** We used Pearson Correlation to test the association between Electrodermal Synchrony Scores and Sensorimotor Synchrony Scores (both are continuous variables, $N = 28$ participants (15 men, 13 women), with an age range of 19–28 (*Mean* = 24.88, *SD* = 2.1 years)).

**Testing the association between romantic attractiveness scores and electrodermal synchrony scores.** We tested whether the individual ability of electrodermal synchrony during the date predicts attraction by calculating the correlation between Electrodermal Synchrony Scores and Romantic Attractiveness Scores. We used Spearman Correlation since the attraction was measured as a rank variable on a scale

of one to five. ($N = 32$ participants (16 men, 16 women), with an age range of 19–28 (*Mean* = 24.8, *SD* = 2.02 years)).

**Testing the association between romantic attractiveness scores and sensorimotor synchrony scores.** We used the Spearman Correlation to test the association between the Romantic Attractiveness Scores and the Sensorimotor Synchrony Scores. The Spearman Correlation is applied since attraction was measured as a categorical variable on a scale of one to five. Forty-four participants (23 men, 21 women), aged 19–28 (*Mean* = 24.76, *SD* = 1.97 years), completed both the speed-date and metronome tasks.

### Reporting summary
Further information on research design is available in the Nature Portfolio Reporting Summary linked to this article.

## Results
### Does increasing physiological synchrony increase attraction, and does increased attraction increase synchrony?
While the correlation between physiological synchrony and romantic attraction has been reported by two studies[27,28], the direction of the relationship between synchrony and attraction is still unknown. We assessed the extent to which increased physiological synchrony leads to increased romantic attraction, and vice versa – the extent to which increased initial attraction leads to elevated synchronization.

### A preregistered online experiment: testing the extent to which manipulating physiological synchrony between an actor and an actress determines their attractiveness ratings by participants
In an online experiment with an experimental design, 144 participants observed a video of a man and a woman actors interacting in either high or low synchrony levels. Based on previous research demonstrating an association between synchrony and attraction[27,28], we hypothesized that increasing synchrony between the actors would make participants rate them as more attractive and more attracted to each other.

We applied a repeated measures MANOVA with one independent variable (level of synchrony, high v. low) and two dependent variables (the total attractiveness of the actors and perceived attraction between the actors). As predicted, inducing synchrony significantly increases attraction scores ($F_{(1,142)} = 6.073$; $p = 0.003$). Compared to the low-synchrony interaction, the high-synchrony interaction increased both the attractiveness scores of the actors ($F_{(1,142)} = 5.034$; $p = 0.026$) and the perceived attraction between the actors ($F_{(1,142)} = 11.92$; $p < 0.001$) (Fig. 3). Assumptions on homogeneity of variances, multivariate normality, and normality of residuals were confirmed by Levene's tests (Actors' attractiveness: $F_{(1,142)} = 2.425$, $p = 0.122$; Actors' attraction to each other: $F_{(1,142)} = 0.443$, $p = 0.507$), the Mardia's test (*Skewness statistic* = 7.1, $p = 0.131$; *Kurtosis statistic* = 0.841, $p = 0.4$), and the Shapiro–Wilk test ($W = 0.993$, $p = 0.248$), respectively.

### Testing the extent to which attraction impacts physiological synchrony
To examine the extent to which initial interest in the partner at the beginning of the date determines the electrodermal synchrony between partners during the date, we tested the association between the initial interest in the partner measured before the partners started to interact, and the electrodermal synchrony during the date.

Multilevel model analysis shows that initial interest in the partner at the beginning of the date is not significantly associated with electrodermal synchrony during the date in both men ($\beta = 0.061$; $p = 0.44$; *95% Confidence Interval* = [−0.091, 0.213]; $N = 64$ dates) and women ($\beta = 0.039$; $p = 0.513$; *95% Confidence Interval* = [−0.076, 0.153]; $N = 64$ dates). Given the null results, we also computed Spearman Correlations and the Jeffreys-Zellener-Siow (JZS) Bayes Factors (BFs), for men (*Spearman r* = 0.105; $p = 0.41$; *95% Confidence Interval* = [−0.126, 0.335]; $BF_{01} = 2.44$; $N = 64$ dates), for

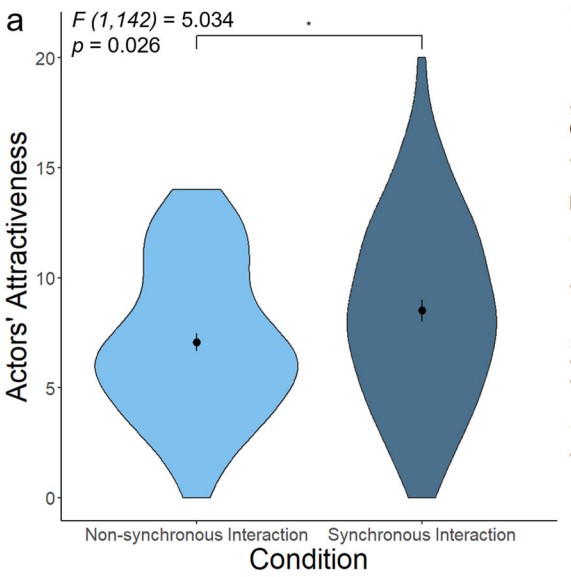

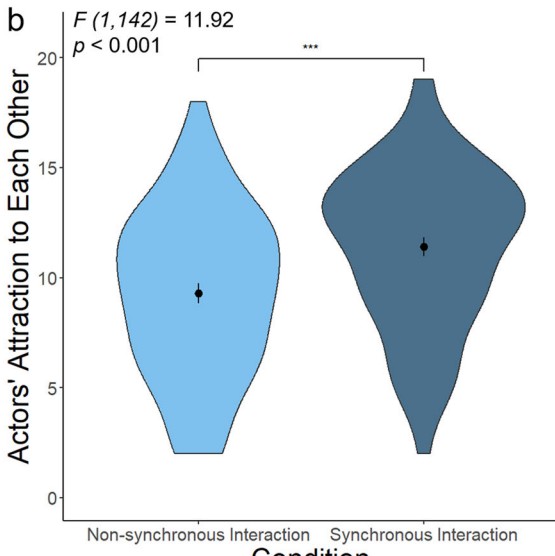

**Fig. 3 | Manipulating synchrony between an actor and an actress determines their perceived attraction by participants. a** An actor and an actress are rated as more attractive when in sync ($M = 8.507$; $SE = 0.493$; *95% Confidence Interval* = [7.542, 9.475]; $N = 73$ participants) compared to the same actor and actress when not in sync ($M = 7.07$; $SE = 0.406$; *95% Confidence Interval* = [6.291, 7.86]; $N = 71$ participants). **b** An actor and an actress are rated as more attracted to each other when in sync

($M = 11.411$; $SE = 0.421$; *95% Confidence Interval* = [10.59, 12.23]; $N = 73$ participants), compared to when not in sync ($M = 9.282$; $SE = 0.451$; *95% Confidence Interval* = [8.404, 10.152]; $N = 71$ participants). The Y-axes represent the sum of ratings to both actors, and the error bars represent 1 standard error from the mean. * represents $p$-values between 0.01 and 0.05; *** represents $p$-values smaller than 0.001.

women (*Spearman* $r = 0.059$; $p = 0.646$; *95% Confidence Interval* = [−0.207, 0.324]; $BF_{01} = 3.162$; $N = 64$ dates), and for both (*Spearman* $r = 0.128$; $p = 0.312$; *95% Confidence Interval* = [−0.129, 0.386]; $BF_{01} = 2.247$; $N = 64$ dates). The results provide anecdotal to moderate support in favor of the null hypothesis, indicating little credible evidence for the effect of initial interest on electrodermal synchrony[100].

Summarizing 1a and 1b demonstrates that the level of synchrony determines the level of attractiveness and that there is no credible evidence that initial interest determines the level of synchrony.

### Is synchrony interrelated across tasks and a determinant of attractiveness?

**Synchrony is correlated across social and nonsocial tasks.** To examine the association between social and nonsocial synchrony, we calculated the correlation between the *Individual Electrodermal Synchrony Scores* (social synchrony during speed-dates) and the *Individual Sensorimotor Synchrony Scores* (nonsocial synchrony in the finger tapping task). Results indicate that individual scores of social and nonsocial synchrony are significantly correlated (*Pearson* $r = 0.494$; $p = 0.008$; *95% Confidence Interval* = [0.148, 0.732]; $N = 28$) (Fig. 4a), as some individuals tend to have increased synchrony scores regardless of the task or the partner, to which we refer as Super Synchronizers (see Supplementary Note S3, Supplementary Figs. S2 and S3 for statistical characterization of *Super Synchronizers*). Pearson Correlation is calculated here since both measurements are continuous.

**Individuals with increased electrodermal synchrony are rated as more attractive.** To examine the extent to which individuals with increased social synchrony are rated as more romantically attractive, we calculated the correlation between the *Individual Electrodermal Synchrony Scores* and the *Individual Romantic Attractiveness Scores*. The results show that the individual propensity to physiologically synchronize with partners while dating is correlated with romantic attractiveness (*Spearman* $r = 0.415$; $p = 0.018$; *95% Confidence Interval* = [0.08, 0.746]; $N = 32$) (Fig. 4b). This effect is optimal when computing the electrodermal synchrony for the first two minutes of the date, based on our previous research showing that this time frame is most predictive of

romantic attraction[28]. When computing the electrodermal synchrony across the entire date, the results are *Spearman* $r = 0.342$; $p = 0.056$; *95% Confidence Interval* = [−0.015, 0.695]; $N = 32$. See Supplementary Note S4 for analyses on the date level.

**Individuals with increased sensorimotor synchrony are rated as more attractive.** To examine the extent to which individuals with better performance in the tapping task are rated as more romantically attractive, we calculated the correlation between the *Individual Sensorimotor Synchrony Scores* and the *Individual Romantic Attractiveness Scores*. The results demonstrate that individuals with an improved ability of sensorimotor synchrony are rated as more attractive (*Spearman* $r = 0.413$; $p = 0.005$; *95% Confidence Interval* = [0.176, 0.648]; $N = 44$) (Fig. 4c).

### Discussion

Our findings demonstrate that experimentally manipulating the physiological synchrony between two partners as they interact affects their perceived attractiveness: the same actors are rated more attractive by participants when in sync than when not in sync. This suggests that attractiveness can be experimentally affected by synchrony. Moreover, the capacity to synchronize is individually associated in social and nonsocial tasks and predicts the participants' attractiveness. Some individuals are *Super Synchronizers*, i.e., demonstrate increased synchrony regardless of the task or the partner, and those individuals are considered more romantically attractive.

The association between electrodermal synchrony and sensorimotor synchrony suggests that social synchrony is rooted in domain-general sensorimotor skills, potentially because both require behavioral adjustment to dynamic sensory input, whether social or nonsocial. Sensorimotor synchrony measured by the finger tapping task requires adjustment of finger movements to an auditory stimulus, and social synchrony measured by electrodermal activity requires adjustment of physiological processes. Notably, both processes are regulated in an anticipatory manner[101–103]. This supports the hypothesis that, like sensorimotor synchrony, social interactions also rely on prediction[104,105], perception-action[106], and learning[106–112], possibly based on general sensorimotor predictive mechanisms needed to interact[73]. Interestingly, improved performance in those features is associated with romantic attraction.

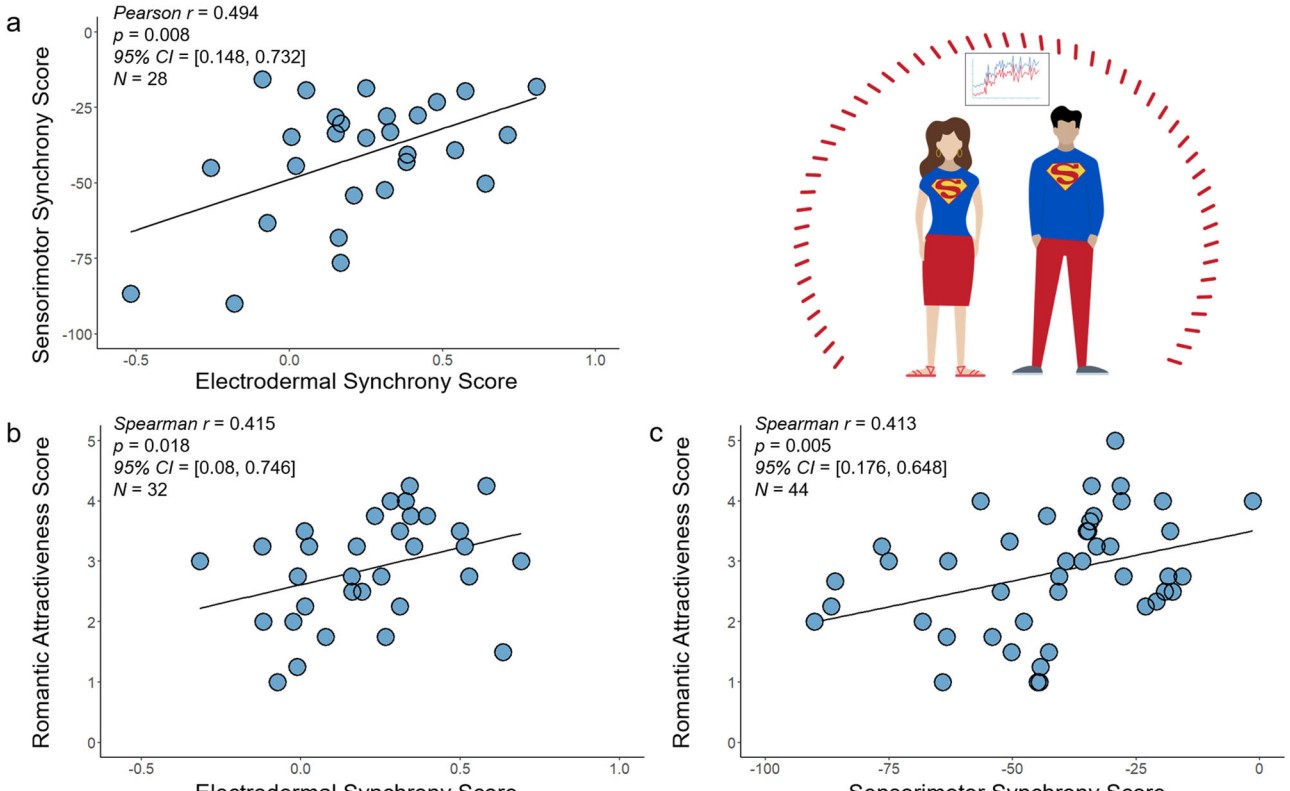

**Fig. 4 | Electrodermal and sensorimotor synchrony are interrelated and attractive. a** The individual ability for synchrony is associated in social and nonsocial tasks: the individual scores of social physiological synchrony with partners while dating correlate with the individual scores of nonsocial sensorimotor synchrony in a finger tapping task. **b** The individual scores of social physiological synchrony with partners while dating correlate with individual scores of romantic attractiveness. **c** Individual scores of nonsocial sensorimotor synchrony correlate with individual scores of romantic attractiveness. Electrodermal synchrony scores were calculated as the average of the electrodermal synchrony scores across four dates. Sensorimotor synchrony scores are calculated by the performance in the finger tapping task, where values closer to zero reflect minimal error and increased sensorimotor synchrony. Romantic attractiveness scores are calculated as the average of the attractiveness ratings across dates.

There are several possible explanations for why synchrony is attractive. First, physiological synchrony can benefit physiological regulation in close partners[28], or- social regulation[113–115]. This hypothesis is supported by evidence showing that physiological synchrony is correlated with improved physiological regulation of different homeostatic systems, including temperature[116], immune function[117], heart rate[64], and affect[113,118]. Through social regulation, humans can learn to prefer synchronizing partners and to synchronize with others to optimize homeostasis[28,30,114,115,119]. Second, interacting with Super Synchronizers, who can better detect and adjust to external cues, can result in a more adapted interaction, with immediate physiological and social gains, making synchronous interactions more rewarding and enjoyable[114,115,120]. Lastly, being a computationally heavy task, the improved ability to synchronize can indicate greater cognitive aptitudes and fitness, which is in line with the 'handicap' principle by Amotz and Avishag Zahavi, suggesting that costly signals reflect the high qualification of their owners[26], which are considered more physiologically fit, and thus more attractive[26]. Importantly, such an evolutionary explanation is a speculation and hard to evaluate empirically. Yet, the hypotheses that synchrony is attractive because it is rewarding and improves physiological regulation are empirical questions that can be addressed in future research.

In addition to physiological synchrony[27,28,36,37,40], behavioral synchrony is another focus of study in romantic attraction[121,122], demonstrating that specific non-verbal behaviors are related to romantic and sexual interest[123–126]. Specifically, increased romantic interest is related to faster and more frequent movements, more patterns initiated by the male, longer patterns of behavior, more repetitions of the same patterns[124,126], and coupling of body sway[127]. In addition to romantic interest and attraction[28,128], coordinated motion was found to be associated with rapport between teacher and student[129], attentiveness in physician-patient interactions[51], and intimacy between same-sex strangers[53]. Such behavioral transactions could serve as a behavioral mechanism that supports physiological synchrony by providing consistent behavioral cues that signal the physiological fluctuations to the partner and enabling an attuned response. Future research is needed to investigate how people regulate their behavior to physiologically synchronize during social interactions, and the extent to which the association between different behavioral patterns and romantic attraction is mediated through social physiological regulation.

## Limitations

Our findings show that the individual capacity for synchrony is associated across social and sensorimotor tasks and predicts attractiveness. This raises the hypothesis that synchrony is an individual trait. While synchrony in this study was associated across social and nonsocial tasks, it was not tested longitudinally. Thus, we cannot yet determine its stability across time. Previous longitudinal research on mothers and children shows that dyadic synchrony is stable across time[130]. Yet, future research is needed to determine whether synchrony is an individual trait, discover its underlying cognitive, behavioral, and neural mechanisms, and assess its social consequences.

Another limitation is the causal inference of synchrony and attraction. This research provides initial evidence that experimentally manipulating synchrony in actors increases their attractiveness ratings by participants. Moreover, increased synchrony during the date precedes increased attraction[27,28], while there is no credible evidence that initial interest at the beginning of the date predicts improved synchrony during the date. This suggests that synchrony potentially elicits attraction, and not the consequence of attraction. However, it is important to note that we can still not

infer the causal effect of synchrony on actual romantic feelings of partners on dates. This must be addressed in future research that manipulates the level of synchrony during actual dates, examining its short-term effects on the mutual romantic interest of the participants, and on the long-term relationship outcomes. Lastly, future research is needed to investigate the role of synchrony in mate selection in homosexuals.

In summary, this research demonstrates that some individuals are Super Synchronizers regardless of the task or the partner, and Super Synchronizers are more attractive. This suggests that mate selection in humans depends on dynamic interactive features that enable humans to attune their physiological regulation.

## Data availability
All data used in the analyses is available at https://osf.io/nzhv6/.

## Code availability
The R code (2022.12.0+353) used for the analyses is available at https://osf.io/nzhv6/.

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

## Acknowledgements

The authors thank Yitzchak Yadegari for the graphic illustrations in Figs. 1 and 4, and Keren Kasten for the graphic illustration in Fig. 2. This study was funded by internal funding from the Hebrew University of Jerusalem granted to Dr. Shir Atzil. The funders had no role in planning the study design, collecting the data, performing the data analyses, publishing, or preparing the manuscript.

## Author contributions

Matan Cohen- conceptualization, methodology, analysis, investigation, data curation, validation, visualization, writing. Maayan Abargil- methodology, validation. Merav Ahissar- conceptualization, methodology, writing. Shir Atzil- conceptualization, methodology, analysis, writing, funding, supervision.

## Competing interests

The authors declare no competing interests.
