## [Peer Review File · Communications Psychology]

18th Oct 23

Dear Dr Atzil,

I hope this message finds you well.

Thank you for your patience during the peer-review process. Your manuscript titled "Synchrony is an individual aptitude that determines romantic attractiveness" has now been seen by 3 reviewers, whose comments are appended below. You will see that they find your work of some potential interest. However, they have raised quite substantial concerns that must be addressed. In light of these comments, we cannot accept the manuscript for publication, but would be interested in considering a revised version that fully addresses these concerns.

We hope you will find the Reviewers' comments useful as you decide how to proceed. Should additional work allow you to address these criticisms, we would be happy to look at a substantially revised manuscript. If you choose to take up this option, please highlight all changes in the manuscript text file, and provide a detailed point-by-point reply to the reviewers.

We understand that given present circumstances, revisions may take much longer than usual. We will be as accommodating as possible, but we would be grateful if you let us know if you require more time, or are unable to resubmit the manuscript.

Editorially, we ask you to address the referees' concerns comprehensively and in particular ensure that the experimental proceedings and analyses (including in relation to the preregistration) are described in greater clarity. It is also important that the results are not over-interpreted; we generally call for caution with regard to causal language, although it is of course appropriate to distinguish between experimental and observational data. In the presentation of Study 1, we recommend that it is made much clearer throughout the Abstract and main text that the perceived synchrony of the targets (actors) affects how participants (raters) rate the targets' attractiveness and attraction towards each other (i.e., the participants and targets are separate groups, there is no evidence from either study for an effect of experimentally controlled synchronicity on romantic feelings between the people displaying synchrony). Similarly, in the absence of repeated measures, the correlation between performance on the tapping task and behavioural synchrony is best described as an association, avoiding language that may suggest evidence for a (stable) trait.

Reviewer #1 raises the point that differences between videos other than synchronicity of the actors may affect the results and Reviewer #2 recommends some additional exploratory analyses of the

visual material. We ask you to include in the revision additional analyses that serve the purpose of ruling out alternative accounts; other exploratory analyses can be included at your discretion.

With regard to the discussion of your results, we recognize that an evolutionary angle is a likely popular explanation; however, as a general principle we ask authors to limit the amount of speculation about untested mechanisms in the Discussion. Conversely, as laid out in our guidelines (please see the template and checklist linked below), we ask you to do include a section under the subheading "Limitations" that forms part of the Discussion where you include a transparent account of limitations of the work, especially those that could not be ruled out on the basis of further analysis.

Likewise following our editorial policies for the reporting and interpretation of statistics, please provide Bayesian statistics or equivalence tests for the null results reported on lines 157-159. Similarly, as we do not permit marginally significant results to be interpreted, please rephrase the sentence on line 217, which extends the statement that an effect has been established to the entire date (period) based on $p = 0.056$.

Please use the following link to submit your revised manuscript, point-by-point response to the Reviewers' comments with a list of your changes to the manuscript text (which should be in a separate document to any cover letter) and any completed checklist:

[link redacted]

Please do not hesitate to contact me if you have any questions or would like to discuss the required revisions further. Thank you for the opportunity to review your work.

Best regards,

Yafeng Pan

Yafeng Pan, PhD

Editorial Board Member

Communications Psychology

orcid.org/0000-0002-5633-8313

EDITORIAL POLICIES AND FORMATTING

Editorial Policy: Policy requirements (Download the link to your computer as a PDF.)

Furthermore, please align your manuscript with our format requirements, which are summarized on the following checklist:

Communications Psychology formatting checklist

and also in our style and formatting guide Communications Psychology formatting guide .

* **CODE AVAILABILITY:** All Communications Psychology manuscripts must include a section titled "Code Availability" at the end of the methods section. In the event of publication, we require that the custom analysis code supporting your conclusions is made available in a publicly accessible repository; please choose a repository that provides a DOI for the code; the link to the repository and the DOI must be included in the Code Availability statement. Publication as Supplementary Information will not suffice. We ask you to prepare and upload code at this stage, to avoid delays later on in the process.

* **DATA AVAILABILITY:**

All Communications Psychology research manuscripts must include a section titled "Data Availability" at the end of the Methods section or main text (if no Methods). More information on this policy, is available at <http://www.nature.com/authors/policies/data/data-availability-statements-data-citations.pdf>.

At a minimum the Data availability statement must explain how the data can be obtained and whether there are any restrictions on data sharing. Communications Psychology strongly endorses open sharing of data. If you do make your data openly available, please include in the statement:

We recommend submitting the data to discipline-specific, community-recognized repositories, where possible and a list of recommended repositories is provided at <http://www.nature.com/sdata/policies/repositories>.

If a community resource is unavailable, data can be submitted to generalist repositories such as figshare or Dryad Digital Repository. Please provide a unique identifier for the data (for example a DOI or a permanent URL) in the data availability statement, if possible. If the repository does not provide identifiers, we encourage authors to supply the search terms that will return the data. For data that have been obtained from publicly available sources, please provide a URL and the specific data product name in the data availability statement. Data with a DOI should be further cited in the methods reference section.

REVIEWER EXPERTISE:

Reviewer #1 synchrony/romantic bonding

Reviewer #2 behavioural/physiological synchrony

Reviewer #3 physiological synchrony/behavioral synchrony, bonding

Reviewer #1 (Remarks to the Author):

In this study, Cohen et al. tried to answer why we are attracted to specific individuals. They hypothesized that synchrony is an individual aptitude that determines romantic attractiveness. They first created two videos with different physiological synchrony and found that the third-party viewers rated the actors in the video with higher physiological synchrony as more attractive. Subsequently, a naturalistic speed dating experiment and finger-tapping tasks were conducted to measure social physiological synchrony and nonsocial sensorimotor synchrony separately. The findings indicated that an individual's propensity to synchronize remains consistent across social and nonsocial tasks. Individuals with greater synchrony were perceived as more attractive.

This research question is both interesting and significant. The objectives and the rationale of the study are clearly stated. The statistical analyses are well described. However, the interpretation of the results appears somewhat overstated, which I would like to go into in more detail below.

1. I wonder whether the synchrony is really the cause of attractiveness. In the online experiment (Results, 1a; lines 114-145), the results showed that videos with varying synchrony levels elicited different attractiveness ratings. Yet, could there be other different elements in the videos, such as the frequency of verbal or nonverbal responses or arousal levels? Is it possible that synchrony is a consequence of attractiveness rather than its cause?
2. I'm a bit confused as to why the statistical analysis of 1b (lines 156-157) used multilevel model analysis, while 2b (lines 211-213) did not. Could the negative result in 1b be attributed to a distinct statistical analysis?
3. The stimuli section in the online experiment warrants a more thorough description. For example, what instructions were given to the actors to induce different levels of synchrony (line 321)? What were the items of the perceived synchrony ratings (line 329)? Were the two videos well-controlled? If they differ in many aspects, might the cause of attractiveness be something other than synchrony?

4. Given concerns about the causal relationship between synchrony and attractiveness, it might be appropriate to tone down the title, abstract, results (lines 160-162), and discussion section (lines 243-249, 273-276) that suggest causality.
5. The authors did not clearly state the limitations of their study. This part should be included in the discussion.
6. Comparisons with other studies need to be expanded in the discussion section.

Reviewer #2 (Remarks to the Author):

This manuscript describes a nicely combined multimethodological approach to the phenomenon of synchrony during interpersonal attraction/interaction. The different aspects of the study are carried out in a straightforward manner, and the provided information regarding the procedures are for the larger part understandable and reasonable.

The study may enrich our understanding of the connections between individual (trait-level) aspects related to synchrony, and how these play out during real (naturalistic) dyadic interactions.

The experiment was preregistered, and the authors follow their planned approach. Additionally, an extended (explorative) analysis of further associations is provided.

I see some room for improvement/additions in the discussion, and I suggest the inclusion of e.g. evolutionary-based theories regarding interpersonal attraction and phenomena of synchrony. A simple (post-hoc) assessment of the video-stimuli used in exp1 could furthermore shed light on possible facets of synchrony-detection/observation in this study.

Generally speaking, the study provides relevant bases for further research and may inform and enrich a wide range of readers.

I hope the following more detailed comments will help the authors in strengthening their work:

Abstract:

Controlling the level of ...  I think the term „controlling“ is somewhat misleading, because in my understanding, it implies a modification (directly influencing the level of synchrony), that was not the case in this study. I suggest to reword this term into something like e.g. „inducing“/„manipulating“ the level of physiological synchrony by means of task characteristics.

Online experiment:

On l. 117: please provide one or two sentences how this was achieved (what was the task?)

l. 138: instead of controlling, I suggest „manipulating“

Results:

I wonder whether using all the data from the encounters would affect the results, i.e. how about not averaging synchrony/attraction across meetings, but instead using a multilevel approach for these repeated measurements (mixed-effects models).  at least in the supplementary/exploratory analysis.

Discussion:

I suggest adding older research on initial attraction, e.g. some information by the work of Grammer and colleagues:

Grammer, K., Kruck, K., Juette, A., & Fink, B. (2000). Non-verbal behavior as courtship signals: The role of control and choice in selecting partners. *Evolution and Human Behavior*, 21(6), 371-390. doi:10.1016/S1090-5138(00)00053-2

Grammer, K., Honda, R., Schmitt, A., & Juette, A. (1999). Fuzziness of nonverbal courtship communication unblurred by motion energy detection. *Journal of Personality and Social Psychology*, 77(3), 487-508. doi:10.1037/0022-3514.77.3.487

Grammer, K., Kruck, K. B., & Magnusson, M. S. (1998). The courtship dance: Patterns of nonverbal synchronization in opposite-sex encounters. *Journal of Nonverbal Behavior*, 22(1), 3-29. doi:10.1023/A:1022986608835

Furthermore, I would like to suggest to more fully discuss possible implications of "individual aptitude".

Methods:

I. 327: Apart from the assessment of physiological synchrony, I suggest to determine the level of movement synchrony in these video-clips. Multiple studies in the domain of psychotherapy and social interaction in general have shown the influence of movement synchrony for relationship quality and task/therapy-outcome. One simple measure for this could be based on frame-differencing, as described in Ramseyer (2020). A more complex analysis would be possible by using e.g. OpenPose (Cao et al., 2021). Using such a visually-based quantification of synchrony would importantly inform this study as to whether the raters in experiment 1 relied more on visual information (if the movement synchrony is different in the 2 conditions), or if their assessment was probably based on other features. I am suggesting this addition, because it can be very easily (and quickly) implemented, and it would shed light on whether the higher physiological synchrony (which was probably not detectable in the video) was „ratable“ because of movement quality or not.

Cao, Z., Hidalgo, G., Simon, T., Wei, S. E., & Sheikh, Y. A. (2021). Openpose: Realtime multi-person 2d pose estimation using part affinity fields. *IEEE Transactions on Pattern Analysis and Machine Intelligence*, 43(1), 172-186. doi:10.1109/TPAMI.2019.2929257

Ramseyer, F. T. (2020). Motion energy analysis (MEA). A primer on the assessment of motion from video. *Journal of Counseling Psychology*, 67(4), 536-549. doi:10.1037/cou0000407

I. 321: Please provide examples for the kind of instructions given to the actors. What did they do? how was this stark difference in physiological synchrony achieved? More information regarding the content, duration, setting should be provided.

Reviewer #3 (Remarks to the Author):

This article presents two interesting studies about physiological synchrony in the context of romantic attractiveness.

Overall, the article is very interesting, and the two experiments seem to be scientifically sound and the proposed methods suitable for the verification of the proposed hypothesis. The employed statistical methods appear correctly applied, and at least for the first experiment, the preregistered plan has been followed. While I believe the article presents a very interesting topic, and the procedures seem appropriate, I believe some work should be done on the manuscript to improve its readability and to clarify the work of the authors. I have detailed my concerns below.

Major Issues

- If I understood correctly, you have two sets of data, one collected online (experiment 1a) and one collected in the lab (1b and 2). This second dataset has been used in full for experiment 2, while only the first round of speed-dates has been used for experiment 1b. Is this correct? If this is the case I believe it should be clarified in the article, as in the current form is not very clear. Moreover, there seems to be a bit of confusion between the reporting of 1b and 2 (e.g. the caption of Figure 1 indicates a total of 48 participants, however in the text the figure is referenced on Line 152, when the authors are describing study 1b, with a total of 32 participants).

- For what concerns experiment 1a, the methods and analysis are clear to me. However, I have one question about the procedure. One may wonder whether all the participants shared a unique definition of “synchrony” (Line 315). Did the authors explain before the experiment (e.g. during the on boarding of the participant) what was synchrony in the context of their study?

- Moving to the EDA acquisition and analysis, I have a couple comments. First, I would suggest the authors to move the paragraph starting at Line 377 (details about the Empatica’s technical specs) earlier in the manuscript (e.g. during experiment 1a, e.g. Line 323). Moreover, I believe more details should be given with respect to the analysis of the signal. Were the signals preprocessed before the measurement of the correlation? If so, what is the employed pipeline and how were signals processed (e.g. software, version)?

- Throughout the manuscript, in multiple lines, the authors refer to “Super Synchronizers”. While the concept is interesting, I believe more details should be reported on this (e.g. what is the prevalence in your sample?, how do their results differ from the results of the other participants? For example, do the results presented in Figure 4 differ between Super Synchronizers and the other participants? I would suggest the authors to put more emphasis on this topic if they are citing it several times along the manuscript.

Minor Issues

- For reproducibility, I would suggest the authors to report the version and operating system on which R and G*Power have been used.

- I am thankful to the authors for pre-registering their study and for following the plan. They really did a great job on this. I have however a point that I believe should be clarified in the article: the authors estimated the number of required participants based on a MANOVA, but then in the manuscript they report using a multivariate GLM (Line 130) to test their hypothesis. I would suggest the authors to clarify the relation between the two models in the methods section.

- There seems to be an extra character (a star) in the title of ref 96, Line 714.

Overall, I believe that the manuscript presents two very interesting studies that together confirm the importance of synchronization on the selection of romantic partners. While I have found the methods and analytic plan suitable for the study of the hypothesis, I believe more work should be done in the reporting, to improve the clarity and quality of the manuscript.

Reviewer 1:

1. I wonder whether the synchrony is really the cause of attractiveness. In the online experiment (Results, 1a; lines114-145), the results showed that videos with varying synchrony levels elicited different attractiveness ratings. Yet, could there be other different elements in the videos, such as the frequency of verbal or nonverbal responses or arousal levels?

We thank the reviewer for raising this point. While preparing the stimuli, we deliberately controlled for elements in the video that could potentially impact the participants' ratings, including the interaction setup, the actors' positions, the conversation content, physical appearance, and the video length.

To further address the Reviewer's point, along with points 2 and 8 of Reviewer 2, the revised manuscript now includes behavioral analyses to characterize the behavioral display of the actors in the two videos. Specifically, we coded the actors' facial expressions, body movements, gaze, and vocalizations using a second-by-second behavioral coding procedure commonly used in our lab (see Abu-Salih et al., 2023 for a detailed method²). The physiological and behavioral differences between the two videos are summarized in Table S1 (from the largest to the lowest effect).

Table S1: Quantifying behavior and physiology of the actor and actress in the synchronous and non-synchronous videos.

Variable	Synchronous video	Non-synchronous video	Ratio Synch/Non-synch (Absolute Value)
Physiological (EDA) man-woman synchrony	0.614	-0.097	6.33
Social gaze woman	0.796	0.323	2.464
Facial expression man-woman synchrony	0.359	0.173	2.075
Social gaze man	0.763	0.527	1.448

Body movement woman	1.108	0.871	1.272
Facial expression man	2.366	2.194	1.078
Vocalization man	1.022	0.957	1.068
Vocalization woman	1.366	1.29	1.059
Facial expression woman	2.355	2.419	0.974
Body movement man	1.183	1.473	0.803
Vocalization man-woman synchrony	-0.312	-0.435	0.717
Body movement man-woman synchrony	0.169	-0.241	0.701
Physiological arousal (EDA) woman	0.388	0.961	0.404
Physiological arousal (EDA) man	0.969	3.083	0.314
Social gaze man-woman synchrony	-0.031	-0.129	0.24

The ratio is calculated on the correlation coefficients' absolute values to account for both the in-phase and anti-phase mutual change over time³.

This post hoc descriptive analysis reveals possible bio-behavioral facets that impact the raters' attraction detection. The analysis shows that the most significant difference between the two videos is the physiological synchrony between the man and woman; it increased six times in the synchronous video compared to the non-synchronous video. This provides a strong manipulation check to the experimental paradigm. Additionally, synchronization in facial expressions is two times higher in the synchronous video than the non-synchronous video. Social gaze was slightly more evident in the synchronous video (2.46 times more social gaze in the woman and 1.45 times

more social gaze in the man in the synchronous interaction). Last, the physiological arousal levels are lower in the synchronous video (3.18 times lower in the man and 2.48 times lower in the woman).

The behavioral differences between the videos can serve two functions in this experiment. The increased synchrony in facial expression and increased gaze by the woman can serve as visual cues for the raters, used to assess the attractiveness of the actors. Second, it can serve the actors to synchronize when instructed. Gaze was previously associated with higher synchrony in different processes, such as neural activity⁴⁻⁶ and behavior⁷. However, increased gaze was tested as a marker of attraction and was not predictive of romantic interest on its own⁸. Likewise, in this study, lower arousal was not associated with attraction on its own (measured in the dating experiment, Spearman $r = -0.12$; $p = 0.343$; 95% confidence interval = $[-0.407, 0.166]$; $N = 64$). Future research is needed to investigate further how people regulate their behavior in order to synchronize during social interactions, and how different behavioral patterns potentially serve as markers for romantic attraction.

These analyses are mentioned in the revised Results Section (page 5), Methods Section (page 15) and fully described and discussed in the Supplementary Results Section S1 along with Table S1 and Figure S1 (pages 1-5 in the Supplementary Results).

2. Is it possible that synchrony is a consequence of attractiveness rather than its cause?

To test for this possibility, in the lab experiment, we measured participants' romantic interest in their partner twice: once at the beginning of the date prior to any interaction, and again after the date. To test whether synchrony is the consequence of initial attraction, we tested the extent to which the initial attraction is predictive of the synchrony during the date. We found no significant effect in both men ($\beta = 0.061$, $p = 0.44$) and women ($\beta = 0.039$, $p = 0.513$). Given the null result, the revised manuscript also includes a Bayesian analysis of this association. This is specified in the Results Section (page 7). The direction of the association between synchrony and attraction is now discussed in the Limitations paragraph of the revised Discussion (pages 12-13).

3. I'm a bit confused as to why the statistical analysis of 1b (lines 156-157) used multilevel model analysis, while 2b (lines 211-213) did not. Could the negative result in 1b be attributed to a distinct statistical analysis?

The difference between the two analyses is that in 1b, the units are 'dates', whereas in 2b, the units are participants. In 1b, since each participant participates in 2-4 dates, some dates in the sample are dependent, and this shared variance is controlled with a multilevel model analysis⁹⁻¹¹, accounting for the random effects for the intercept of recurring data from individuals who participated on multiple dates. To further address the reviewer's comment, we performed a Spearman correlation in 1b, also leading to a null result (*Spearman* $r = 0.059$, $p = 0.646$, 95% *confidence interval* = [-0.207, 0.324]). To further solidify the null result in 1b, and in line with comment 5 of the editor, the revised Results Section now includes a Bayesian analysis that supports the null hypothesis about the association between initial interest and electrodermal synchrony during the date (page 7).

4. The stimuli section in the online experiment warrants a more thorough description. For example, what instructions were given to the actors to induce different levels of synchrony (line 321)? What were the items of the perceived synchrony ratings (line 329)? Were the two videos well-controlled? If they differ in many aspects, might the cause of attractiveness be something other than synchrony?

We thank the reviewer for pointing out the need for further clarification. The revised Methods, Results and Supplementary Sections now include a more detailed description of the instructions (pages 5 and 14), the parameters that were controlled in the two videos (pages 5 and 14-15), and a new analysis of the behavioral differences in the videos of the online experiment (pages 1-5 in Supplementary Results).

Specifically, to create the high synchrony video, we instructed the actors to attune to their partners, whereas for the low synchrony video, we instructed the actors to act independently of their partners. Importantly, the videos were well-controlled in the content of the conversation, the physical appearance of the actors, the setup (a dedicated room at our lab with a homey arrangement), and the length of the videos (92 seconds). We confirmed the synchronization between the partners in the stimuli with physiological assessment during the videos.

To evaluate discrete differences between the videos, the revised manuscript now includes behavioral analyses to characterize the behavioral display of the actors in the two videos. Specifically, we coded the actors' facial expressions, body movements, gaze, and vocalizations using a second-by-second behavioral coding procedure commonly used in our lab (see Abu-Salih et al., 2023 for a detailed method²). The physiological and behavioral differences between the two videos are summarized in Table S1 (from the largest to the lowest effect), showing that the most prominent difference between the videos is the physiological differences. Additional mild differences were found, and are discussed in the Supplementary Results Section S1 (pages 1-5 in Supplementary Results).

5. Given concerns about the causal relationship between synchrony and attractiveness, it might be appropriate to tone down the title, abstract, results (lines 160-162), and discussion section (lines 243-249, 273-276) that suggest causality.

The claim for causality is toned down throughout the manuscript.

6. The authors did not clearly state the limitations of their study. This part should be included in the discussion.

The Discussion Section now includes a Limitations paragraph which clearly states the limitations of this study (pages 12-13).

7. Comparisons with other studies need to be expanded in the discussion section.

The revised discussion now includes comparisons with other studies on synchrony and attraction (page 12).

Reviewer 2:

1. I see some room for improvement/additions in the discussion, and I suggest the inclusion of e.g. evolutionary-based theories regarding interpersonal attraction and phenomena of synchrony.

The revised Discussion now includes a paragraph on interpersonal attraction and synchrony (page 12).

2. A simple (post-hoc) assessment of the video-stiumli used in exp1 could furthermore shed light on possible facets of synchrony-detection/observation in this study.

We thank the reviewer for raising this point. While preparing the stimuli, we deliberately controlled for elements in the video that could potentially impact the participants' ratings, including the interaction setup, the actors' positions, the conversation content, physical appearance, and the video length.

To further address the Reviewer's point, along with point 1 of Reviewer 1, the revised manuscript now includes behavioral analyses to characterize the behavioral display of the actors in the two videos. Specifically, we coded the actors' facial expressions, body movements, gaze, and vocalizations using a second-by-second behavioral coding procedure commonly used in our lab (see Abu-Salih et al., 2023 for a detailed method²). The physiological and behavioral differences between the two videos are summarized in Table S1 (from largest to lowest effect).

Table S1: Quantifying behavior and physiology of the actor and actress in the synchronous and non-synchronous videos.

Variable	Synchronous video	Non-synchronous video	Ratio Synch/Nonsynch (Absolute Value)
Physiological (EDA) man-woman synchrony	0.614	-0.097	6.33
Social gaze woman	0.796	0.323	2.464
Facial expression man-woman synchrony	0.359	0.173	2.075
Social gaze man	0.763	0.527	1.448
Body movement woman	1.108	0.871	1.272
Facial expression man	2.366	2.194	1.078

Vocalization man	1.022	0.957	1.068
Vocalization woman	1.366	1.29	1.059
Facial expression woman	2.355	2.419	0.974
Body movement man	1.183	1.473	0.803
Vocalization man-woman synchrony	-0.312	-0.435	0.717
Body movement man-woman synchrony	0.169	-0.241	0.701
Physiological arousal (EDA) woman	0.388	0.961	0.404
Physiological arousal (EDA) man	0.969	3.083	0.314
Social gaze man-woman synchrony	-0.031	-0.129	0.24

The ratio is calculated on the correlation coefficients' absolute values to account for both the in-phase and anti-phase mutual change over time³.

This post hoc descriptive analysis reveals possible bio-behavioral facets that impact the raters' attraction detection. The analysis shows that the most significant difference between the two videos is the physiological synchrony between the man and woman; it increased six times in the synchronous video compared to the non-synchronous video. This provides a strong manipulation check to the experimental paradigm. Additionally, synchronization in facial expressions is two times higher in the synchronous video than the non-synchronous video. Social gaze was slightly more evident in the synchronous video (2.46 times more social gaze in the woman and 1.45 times more social gaze in the man in the synchronous interaction). Last, the physiological arousal levels are lower in the synchronous video (3.18 times lower in the man and 2.48 times lower in the woman).

The behavioral differences between the videos can serve two functions in this experiment. The increased synchrony in facial expression and increased gaze by the woman can serve as visual cues for the raters, used to assess the attractiveness of the actors. Second, it can serve the actors to synchronize when instructed. Gaze was previously associated with higher synchrony in different processes, such as neural activity⁴⁻⁶ and behavior⁷. However, increased gaze was tested as a marker of attraction and was not predictive of romantic interest on its own⁸. Likewise, in this study, lower arousal was not associated with attraction on its own (measured in the dating experiment, Spearman $r = -0.12$; $p = 0.343$; 95% confidence interval = $[-0.407, 0.166]$; $N = 64$). Future research is needed to investigate further how people regulate their behavior in order to synchronize during social interactions, and how different behavioral patterns potentially serve as markers for romantic attraction.

These analyses are mentioned in the revised Results Section (page 5), Methods Section (page 15) and fully described and discussed in the Supplementary Results Section S1 along with Table S1 and Figure S1 (pages 1-5 in the Supplementary Results).

3. Abstract: Controlling the level of ...  I think the term „controlling“ is somewhat misleading, because in my understanding, it implies a modification (directly influencing the level of synchrony), that was not the case in this study. I suggest to reword this term into something like e.g. „inducing“/„manipulating“ the level of physiological synchrony by means of task characteristics.

In the revised abstract and throughout the manuscript, we now replace “controlling” with “inducing” or “manipulating” (pages 2, 3, 5, 6, and 11).

4. Online experiment: On l. 117: please provide one or two sentences how this was achieved (what was the task?). l. 138: instead of controlling, I suggest „manipulating“

We thank the reviewer for pointing out the need for further clarification. The revised Methods, Results, and Supplementary Sections now include a more detailed description of the instructions (pages 5 and 14), the parameters that were controlled in the two videos (pages 5 and 14-15), and a new analysis of the behavioral differences in the videos of the online experiment (pages 1-5 in Supplementary Results).

Specifically, to create the high synchrony video, we instructed the actors to attune to their partners, whereas for the low synchrony video, we instructed the actors to act independently of their partners. Importantly, the videos were well-controlled in the content of the conversation, the physical appearance of the actors, the setup (a dedicated room at our lab with a homey arrangement), and the length of the videos (92 seconds). We confirmed the synchronization between the partners in the stimuli with physiological assessment during the videos.

‘Controlling’ is revised to ‘manipulating’ (line 140).

5. Results: I wonder whether using all the data from the encounters would affect the results, i.e. how about not averaging synchrony/attraction across meetings, but instead using a multilevel approach for these repeated measurements (mixed-effects models).  at least in the supplementary/exploratory analysis.

As the reviewer suggested, we also conducted a multilevel analysis to assess the association between synchrony and attraction. The electrodermal synchrony during the dates was applied as a fixed effect, while accounting for the random effects for the intercept of recurring data from individuals that participated on multiple dates and for their nesting in specific runs. The models did not converge, indicating a low amount of data for such analysis, which requires a larger sample given the strict control of shared variance. The Spearman correlation between electrodermal synchrony during the entire date and the mutual attraction ratings at the end of the date is Spearman $r = 0.231$, $p = 0.066$, 95% confidence interval = $[-0.003, 0.464]$, $N = 64$ dates. The revised Supplementary Results Section S3 now includes this analysis (pages 7-8 in the Supplementary Results).

6. Discussion:

I suggest adding older research on initial attraction, e.g. some information by the work of Grammer and colleagues:

Grammer, K., Kruck, K., Juetten, A., & Fink, B. (2000). Non-verbal behavior as courtship signals: The role of control and choice in selecting partners. *Evolution and Human Behavior*, 21(6), 371-390. doi:10.1016/S1090-5138(00)00053-2

Grammer, K., Honda, R., Schmitt, A., & Juetten, A. (1999). Fuzziness of nonverbal courtship communication unblurred by motion energy detection. *Journal of Personality and Social Psychology*, 77(3), 487-508. doi:10.1037/0022-3514.77.3.487

Grammer, K., Kruck, K. B., & Magnusson, M. S. (1998). The courtship dance: Patterns of nonverbal synchronization in opposite-sex encounters. *Journal of Nonverbal Behavior*, 22(1), 3-29. doi:10.1023/A:1022986608835

We thank the reviewers for pointing out this important literature. The revised Discussion Section now discusses the non-verbal behaviors related to romantic and sexual interest (page 12).

7. Furthermore, I would like to suggest to more fully discuss possible implications of "individual aptitude".

The revised Discussion Section now discusses the implications of Super Synchrony as an individual aptitude (pages 11-12).

8. Methods:

1. 327: Apart from the assessment of physiological synchrony, I suggest to determine the level of movement synchrony in these video-clips. Multiple studies in the domain of psychotherapy and social interaction in general have shown the influence of movement synchrony for relationship quality and task/therapy-outcome. One simple measure for this could be based on frame-differencing, as described in Ramseyer (2020). A more complex analysis would be possible by using e.g. OpenPose (Cao et al., 2021). Using such a visually-based quantification of synchrony would importantly inform this study as to whether the raters in experiment 1 relied more on visual information (if the movement synchrony is different in the 2 conditions), or if their assessment was probably based on other features. I am suggesting this addition, because it can be very easily (and quickly) implemented, and it would shed light on whether the higher physiological synchrony (which was probably not detectable in the video) was „ratable“ because of movement quality or not.

Cao, Z., Hidalgo, G., Simon, T., Wei, S. E., & Sheikh, Y. A. (2021). Openpose: Realtime multi-person 2d pose estimation using part affinity fields. *IEEE Transactions on Pattern Analysis and Machine Intelligence*, 43(1), 172-186. doi:10.1109/TPAMI.2019.2929257

Ramseyer, F. T. (2020). Motion energy analysis (MEA). A primer on the assessment of motion from video. *Journal of Counseling Psychology*, 67(4), 536-549. doi:10.1037/cou0000407

We thank the reviewer for this comment. To address the reviewer's comment, the revised manuscript now includes a behavioral analysis of the synchronous and nonsynchronous videos in the online experiment, including the extent of movement using a second-by-second behavioral coding procedure commonly used in our lab (see Abu-Salih et al., 2023 for a detailed method²) (pages 1-5 in the Supplementary Results). Please also see the answer to point 2 for a detailed description of this analysis.

9. 1. 321: Please provide examples for the kind of instructions given to the actors. What did they do? How was this stark difference in physiological synchrony achieved? More information regarding the content, duration, setting should be provided.

The revised Methods, Results, and Supplementary Sections now include a more detailed description of the instructions (pages 5 and 14), the parameters that were controlled in the two videos (pages 5 and 14-15), and a new analysis of the behavioral differences in the videos of the online experiment (pages 1-5 in the Supplementary Results). Please see our answer to point 4 for a more detailed description.

Reviewer 3:

1. If I understood correctly, you have two sets of data, one collected online (experiment 1a) and one collected in the lab (1b and 2). This second dataset has been used in full for experiment 2, while only the first round of speed-dates has been used for experiment 1b. Is this correct? If this is the case I believe it should be clarified in the article, as in the current form is not very clear.

We thank the reviewer for pointing out the need for clarification. For analysis 1b, only rounds 1-4 are used since it analyzes physiological data, which is only available for these rounds. The revised Results Section now clarifies this and specifies the number of participants included in each experiment (pages 6-7).

This study includes two separate data sets: an online experiment with 144 participants who rated the attractiveness of actors in a synchronous and a nonsynchronous video, and a lab-based speed-dating experiment with 48 participants who performed a sensorimotor synchronization task, as well as participated on 85 dates across 7 runs. Due to a technical problem in the physiological measures during runs 5-7 (16 participants), the physiological data was properly collected only during runs 1-4 (32 participants). Due to an equipment malfunction in the finger tapping task, the data of 4 participants is missing. As a result, physiological data is reported for 32 participants out of the 48; the attractiveness ratings are reported for all 48 participants; the sensorimotor performance is reported for 44 participants out of the 48.

2. Moreover, there seems to be a bit of confusion between the reporting of 1b and 2 (e.g. the caption of Figure 1 indicates a total of 48 participants, however in the text the figure is referenced on Line 152, when the authors are describing study 1b, with a total of 32 participants).

In the revised manuscript, both the legend of Figure 1 (page 4) and the text that describes Study 1b (pages 6-7) now report the total number of participants (48). The relevant analyses in the revised Results Section specify that physiological measures are missing for some participants (pages 6-7).

3. For what concerns experiment 1a, the methods and analysis are clear to me. However, I have one question about the procedure. One may wonder whether all the participants shared a unique definition of “synchrony” (Line 315). Did the authors explain before the experiment (e.g. during the on boarding of the participant) what was synchrony in the context of their study?

At the end of the experiment, participants were asked to rate the perceived behavioral synchrony on the date on a Likert scale (0 is the lowest score, 10 is the highest score), as a manipulation validation. We intentionally did not introduce the concept of synchrony before the experiment, in

order not to prime participants while rating the attractiveness of the actors. This is now specified in the Methods Section (page 14).

4. Moving to the EDA acquisition and analysis, I have a couple comments. First, I would suggest the authors to move the paragraph starting at Line 377 (details about the Empatica's technical specs) earlier in the manuscript (e.g. during experiment 1a, e.g. Line 323).

This paragraph has now been put forward and included in Experiment 1a (page 15).

5. Moreover, I believe more details should be given with respect to the analysis of the signal. Were the signals preprocessed before the measurement of the correlation? If so, what is the employed pipeline and how were signals processed (e.g. software, version)?

We thank the reviewer for pointing out the need to clarify. The revised Methods Section now includes more details about the analysis of the electrodermal signal (pages 15-16).

Electrodermal activity (EDA) is controlled by the sympathetic nervous system and reflects the level of physiological arousal^{12,13}. EDA refers to the continuous variation in the electrical characteristics of the skin. Varying numbers of eccrine sweat glands secrete varying amounts of sweat depending on the degree of sympathetic activation. As more sweat is secreted, electrodermal activity increases^{12,14,15}. Typically, electrodermal activity is measured as skin conductance by applying a small, constant voltage to the skin. Skin conductance can be calculated by measuring the current flow through the electrodes, as the voltage is kept constant^{14,16,17}.

The wristbands contain an electrodermal activity sensor with a sampling frequency of 4 Hz, resolution of one digit –900 pSiemens, range of 0.01–100 μ Siemens, and alternating current (8 Hz frequency) with a maximum peak to peak value of 100 μ Amps (at 100 μ Siemens)¹⁸. The wristbands were placed on the wrist of the left hand and recorded the electrodermal signal throughout the interaction. After each run, the obtained signal was uploaded to the E4 application for Windows, and then downloaded from the Empatica website. The preprocessing of the raw data included the temporal alignment of the data from both partners based on a global timestamp^{18,19}. Then, to calculate the electrodermal synchrony between the partners, we applied Pearson correlation using R (2022.12.0+353)⁹.

6. Throughout the manuscript, in multiple lines, the authors refer to “Super Synchronizers”. While the concept is interesting, I believe more details should be reported on this (e.g. what is the prevalence in your sample?, how do their results differ from the results of the other participants? For example, do the results presented in Figure 4 differ between Super Synchronizers and the other participants? I would suggest the authors to put more emphasis on this topic if they are citing it several times along the manuscript.

We thank the reviewer for these points. To address this comment, the revised manuscript now includes supplementary results that specify the experimental definition and characterization of super synchronizers (Supplementary Figures S2 and S3, pages 5-7 in the Supplementary Results):

The distribution of individual electrodermal synchrony scores across all participants ranged from -0.316 to 0.692. Super Synchronizers were defined as the upper third of this distribution [range: 0.33, 0.692], whereas Medium Synchronizers were defined as the middle third [range: 0.16, 0.311], and Low Synchronizers as the lower third [range: -0.316, 0.078]. Accordingly, the difference between the groups captures the variation in synchrony scores: $F(2, 29) = 70.36$, $p\text{-value} < 0.001$, $N = 32$ (Low Synchronizers vs. Medium Synchronizers: $p\text{-value} < 0.001$, Cohen's $d = 3.233$, $df = 15.873$, $t = -7.6$; Low Synchronizers vs. Super Synchronizers: $p\text{-value} < 0.001$, Cohen's $d = 4.427$, $df = 19.367$, $t = -10.381$; Medium Synchronizers vs. Super Synchronizers: $p\text{-value} < 0.001$, Cohen's $d = 2.358$, $df = 14.351$, $t = -5.574$) (Figure S2a).

When assessing the associations between social synchrony, nonsocial synchrony, and attractiveness separately for the different synchrony groups (Figure S3), we find that low social synchronizers (low social physiological synchrony) are significantly less synchronized in the nonsocial sensorimotor task than medium and super synchronizers: $F(2, 25) = 6.658$, $p\text{-value} = 0.005$, $N = 28$ (Low Synchronizers vs. Medium Synchronizers: $p\text{-value} = 0.022$, Cohen's $d = 1.261$, $df = 11.453$, $t = -2.654$; Low Synchronizers vs. Super Synchronizers: $p\text{-value} = 0.016$, Cohen's $d = 1.353$, $df = 10.766$, $t = -2.87$) (Figure S2b). When assessing the association between synchrony and attraction, we find that super social synchronizers (high social physiological synchrony) are rated as more attractive than low synchronizers: $F(2, 29) = 4.64$, $p\text{-value} = 0.018$, $N = 32$ (Low Synchronizers vs. Super Synchronizers: $p\text{-value} = 0.008$, Cohen's $d = 1.246$, $df = 19.916$, $t = -2.923$) (Figure S2c).

Figure S2. Synchrony and attraction in the different synchrony categories. The lower and upper hinges correspond to the 25th and 75th percentiles, while the lower and upper whiskers extend to the smallest and largest values (data points beyond the end of the whiskers are outliers). * represents p-values between 0.01 and 0.05; *** represents p-values smaller than 0.001.

Figure S3. The relationships between electrodermal synchrony, sensorimotor synchrony, and romantic attractiveness in the different synchrony groups.

Minor Issues:

7. For reproducibility, I would suggest the authors to report the version and operating system on which R and G*Power have been used.

The versions are now reported in the revised Methods Section (pages 13 and 16). Moreover, the scripts and data are uploaded to osf.org.

8. I am thankful to the authors for pre-registering their study and for following the plan. They really did a great job on this.

Thank you!

9. I have however a point that I believe should be clarified in the article: the authors estimated the number of required participants based on a MANOVA, but then in the manuscript they report using a multivariate GLM (Line 130) to test their hypothesis. I would suggest the authors to clarify the relation between the two models in the methods section.

We thank the reviewer for pointing out the need for clarification. The preregistered and applied test is repeated measures MANOVA, between two different conditions (synchronous video; non-synchronous video), with one independent variable (bio-behavioral synchrony), and two dependent variables (attractiveness of the actors, attraction between the actors). We used MANOVA since it is the most appropriate multivariate GLM analysis for our data. The revised manuscript now applies a unified terminology for this test (pages 5 and 13).

10. There seems to be an extra character (a star) in the title of ref 96, Line 714.

Corrected, thank you.

References

1. Feldman, R. The relational basis of adolescent adjustment: trajectories of mother–child interactive behaviors from infancy to adolescence shape adolescents’ adaptation. *Attachment & Human Development* **12**, 173–192 (2010).
2. Abu Salih, M. *et al.* Evidence for cultural differences in affect during mother–infant interactions. *Sci Rep* **13**, 4831 (2023).
3. Miles, L. K., Nind, L. K. & Macrae, C. N. The rhythm of rapport: Interpersonal synchrony and social perception. *Journal of Experimental Social Psychology* **45**, 585–589 (2009).
4. Hirsch, J., Zhang, X., Noah, J. A. & Ono, Y. Frontal temporal and parietal systems synchronize within and across brains during live eye-to-eye contact. *NeuroImage* **157**, 314–330 (2017).

5. Gumilar, I. *et al.* Inter-brain Synchrony and Eye Gaze Direction During Collaboration in VR. in *CHI Conference on Human Factors in Computing Systems Extended Abstracts* 1–7 (ACM, New Orleans LA USA, 2022). doi:10.1145/3491101.3519746.
6. Kinreich, S., Djalovski, A., Kraus, L., Louzoun, Y. & Feldman, R. Brain-to-Brain Synchrony during Naturalistic Social Interactions. *Sci Rep* **7**, 17060 (2017).
7. Cappella, J. N. Behavioral and judged coordination in adult informal social interactions: Vocal and kinesic indicators. *Journal of Personality and Social Psychology* **72**, 119–131 (1997).
8. Grammer, K., Honda, M., Jutte, A. & Schmitt, A. Fuzziness of nonverbal courtship communication unblurred by motion energy detection. *Journal of Personality and Social Psychology* **77**, 487–508 (1999).
9. Zeevi, L. *et al.* Bio-behavioral synchrony is a potential mechanism for mate selection in humans. *Sci Rep* **12**, 4786 (2022).
10. Ackerman, R. A., Kashy, D. A. & Corretti, C. A. A tutorial on analyzing data from speed-dating studies with heterosexual dyads: Analyzing speed-dating data. *Pers Relationship* **22**, 92–110 (2015).
11. Kenny, D. A. & Kashy, D. A. Dyadic data analysis using multilevel modeling. in *Handbook of advanced multilevel analysis* 343–378 (Routledge, 2011).
12. Dawson, M. E., Schell, A. M. & Filion, D. L. The electrodermal system. (2017).
13. Wallin, B. G. Sympathetic Nerve Activity Underlying Electrodermal and Cardiovascular Reactions in Man. *Psychophysiology* **18**, 470–476 (1981).
14. Mendes, W. B. Assessing autonomic nervous system activity. *Methods in social neuroscience* **118**, 21 (2009).

15. Quigley, K. S. Sympathetic Nervous System. in *The Corsini Encyclopedia of Psychology* (eds. Weiner, I. B. & Craighead, W. E.) 1–2 (Wiley, 2010). doi:10.1002/9780470479216.corpsy0969.
16. Fowles, D. C. *et al.* Publication Recommendations for Electrodermal Measurements. *Psychophysiology* **18**, 232–239 (1981).
17. Lykken, D. T. & Venables, P. H. DIRECT MEASUREMENT OF SKIN CONDUCTANCE: A PROPOSAL FOR STANDARDIZATION. *Psychophysiology* **8**, 656–672 (1971).
18. UM-16Rev.2.0 20201020-E4_user_manual.pdf | Powered by Box. <https://empatica.app.box.com/v/E4-User-Manual>.
19. E4_gettingstrated.pdf | Powered by Box. <https://empatica.app.box.com/v/E4-getting-started>.

2nd May 24

Dear Dr Atzil,

Your manuscript titled "Social and nonsocial synchrony are interrelated and romantically attractive" has now been seen by our reviewers, whose comments appear below. In light of their advice I am delighted to say that we are happy, in principle, to publish a suitably revised version in Communications Psychology under the open access CC BY license (Creative Commons Attribution v4.0 International License).

We therefore invite you to revise your paper one last time to address the remaining concerns of our reviewers and a list of editorial requests. At the same time we ask that you edit your manuscript to comply with our format requirements and to maximise the accessibility and therefore the impact of your work.

Editorially, we consider that you address the R3's remaining comment, that is, please clarify how you measured participants' romantic interest in the Procedure section.

EDITORIAL REQUESTS:

SUBMISSION INFORMATION:

OPEN ACCESS:

Communications Psychology is a fully open access journal. Articles are made freely accessible on publication under a CC BY license (Creative Commons Attribution 4.0 International License). This license allows maximum dissemination and re-use of open access materials and is preferred by many research funding bodies.

For further information about article processing charges, open access funding, and advice and support from Nature Research, please visit <https://www.nature.com/commspsychol/article-processing-charges>

At acceptance, you will be provided with instructions for completing this CC BY license on behalf of all authors. This grants us the necessary permissions to publish your paper. Additionally, you will be asked to declare that all required third party permissions have been obtained, and to provide billing information in order to pay the article-processing charge (APC).

* **DATA AVAILABILITY:**

[link redacted]

Best regards,

Jennifer Bellingtier

Jennifer Bellingtier, PhD

Senior Editor

Communications Psychology

Yafeng Pan, PhD

Editorial Board Member

Communications Psychology

orcid.org/0000-0002-5633-8313

REVIEWERS' EXPERTISE:

Reviewer #1 synchrony/romantic bonding

Reviewer #2 behavioural/physiological synchrony

Reviewer #3 physiological synchrony/behavioral synchrony, bonding

REVIEWERS' COMMENTS:

Reviewer #1 (Remarks to the Author):

The authors have addressed my concerns with their revision. I have no further comments.

Reviewer #2 (Remarks to the Author):

The authors have succeeded in addressing all points raised in my comments. I am very happy by the way these issues have been both integrated into the manuscript as well as extended in the supplementary material.

The manuscript has gained in clarity, and I have no further issues that need additional attention.

Congratulations on this revision, which was organized in a way that made it easy to follow and evaluate.

Reviewer #3 (Remarks to the Author):

I am thankful to the authors for taking mine and other reviewers comments into consideration. I have found the manuscript to be greatly improved from its last iteration.

I only have one minor note on the work. In your revision letter you mentioned that participants' romantic interest in their partner has been measured twice, before and after the date. However, this doesn't seem to be reflected in the Procedure section (motivation to succeed seems to me different from romantic interest). I think this point should be clarified.